

# Utilization of ripe coconut water in the development of probiotic gelatin

Beatriz Patricio Rocha[1,2], Pedro Luan de Brito Lopes[3], Miqueas Oliveira Morais da Silva[1,2], Ana Catarina Guimarães Gomes[1,2], Flávia Carolina Alonso Buriti[1,2], Isanna Menezes Florêncio[1] and Eliane Rolim Florentino[1,2]

[1] Núcleo de Pesquisa e Extensão em Alimentos, Universidade Estadual da Paraíba, Campina Grande, PB, Brazil
[2] Programa de Pós-Graduação em Ciências Farmacêuticas, Universidade Estadual da Paraíba, Campina Grande, PB, Brazil
[3] Departamento de Química Industrial, Universidade Estadual da Paraíba, Campina Grande, PB, Brazil

Corresponding author
Beatriz Patricio Rocha,
beatrizpattricio@gmail.com

## ABSTRACT

**Background:** Desserts with vegetable ingredients are a constantly expanding global market due to the search for alternatives to cow's milk. Fermentation of these matrices by lactic acid bacteria can add greater functionality to the product, improving its nutritional, sensory, and food safety characteristics, as well as creating bioactive components with beneficial effects on health. Concern for health and well-being has aroused interest in byproducts of the industry that have functional properties for the body, such as mature coconut water, a normally discarded residue that is rich in nutrients. This study aimed to develop a probiotic gelatin based on pulp and water from mature coconuts and evaluate the physicochemical characteristics, viability of the *Lacticaseibacillus rhamnosus* LR32 strain in the medium, as well as the texture properties of the product.

**Methods:** After collection and cleaning, the physicochemical characterization, mineral analysis, analysis of the total phenolic content and antioxidant activity of mature coconut water were carried out, as well as the centesimal composition of its pulp. Afterwards, the gelling was developed with the addition of modified corn starch, gelatin, sucrose, and probiotic culture, being subjected to acidity analysis, texture profile and cell count, on the first day and every 7 days during 21 days of storage, under refrigeration at 5 °C. An analysis of the centesimal composition was also carried out.

**Results:** The main minerals in coconut water were potassium (1,932.57 mg L$^{-1}$), sodium (19.57 mg L$^{-1}$), magnesium (85.13 mg L$^{-1}$) calcium (279.93 mg L$^{-1}$) and phosphorus (11.17 mg L$^{-1}$), while the pulp had potassium (35.96 g kg$^{-1}$), sodium (0.97 g kg$^{-1}$), magnesium (2.18 g kg$^{-1}$), 37 calcium (1.64 g kg$^{-1}$), and phosphorus (3.32 g kg$^{-1}$). The phenolic content of the water and pulp was 5.72 and 9.77 mg gallic acid equivalent (GAE) 100 g$^{-1}$, respectively, and the antioxidant capacity was 1.67 and 0.98 39 g of 2, 2-diphenyl-1-picrylhydrazyl (DPPH) mg$^{-1}$, respectively. The coconut pulp had 2.81 g 100 g$^{-1}$ of protein, 1.11 g 100 g$^{-1}$ of 40 ash, 53% moisture, and 5.81 g 100 g$^{-1}$ of carbohydrates. The gelatin produced during the storage period presented firmness parameters ranging from 145.82 to 206.81 grams-force (gf), adhesiveness from 692.85 to 1,028.63 gf sec, cohesiveness from 0.604 to
0.473, elasticity from 0.901 to 0.881, gumminess from 86.27 to 97.87 gf, and chewiness from 77.72 to 91.98 gf. Regarding the viability of the probiotic microorganism, the dessert had 7.49 log CFU g$^{-1}$ that remained viable during the 21-day storage, reaching 8.51 CFU g$^{-1}$. Acidity ranged from 0.15 to 0.64 g of lactic acid 100 g$^{-1}$. The centesimal composition of the product showed 4.88 g 100 g$^{-1}$ of protein, 0.54 g 100 g$^{-1}$ of ash, 85.21% moisture, and 5.37g 100 g$^{-1}$ of carbohydrates. The development of the gelatin made it possible to obtain a differentiated product, contributing to diversification in the food sector, providing a viable alternative for maintaining consumer health and reducing costs compared to desserts already available on the market.

## INTRODUCTION

The market for fruit and vegetable-based food products has shown significant growth in several countries (*Väkeväinen et al., 2020*). In 2019, global sales of these food products reached values of approximately 55 billion dollars, with the prospect of 60 billion dollars by 2023 (*Conway, 2019*). According to *Euromonitor International (2019)*, about 20% of consumers in the main world markets declared the intention to increase the consumption of plant-based foods and reduce that of animal-based foods. Coconut (*Cocos nucifera* L.) is an important fruit tree found in tropical regions and its fruits can be transformed into various food and beverage products. Typically, people in tropical countries drink coconut water as a sports drink (*Lee, Boo & Liu, 2013*; *Cappelletti et al., 2015*). Coconut water is the clear and nutritious liquid obtained from the coconut endosperm. It is classified as green or mature based on the harvesting season (*Zhang et al., 2018*). Coconut pulp is defined as the product obtained from the edible part of the coconut, through an appropriate technological process. Formed from the fifth or sixth month after the inflorescence, it is a gelatinous and translucent substance that at the end of ripening has a firmer consistency and darker color (*Rocha, Ferreira & Garcia, 2022*). However, mature coconut water is typically discarded. In Thailand, the coconut milk industry directly dumped about 200,000 tons/year into the drain in 2007 (*Unagul et al., 2007*). When discarded without any prior treatment, it becomes a pollutant due to its high biochemical oxygen demand (BOD). Today, the industrial sector's concern is focused on minimizing waste generation, as this is a preventive management system that aims to reduce losses in the production process and improve environmental performance (*Badilla et al., 2020*). Several research groups have focused on the study of coconut water as a popular beverage and sports drink due to its high mineral content and sweet taste (*Lee, Boo & Liu, 2013*; *Cappelletti et al., 2015*; *Prado et al., 2015*). Coconut water contains sugars, vitamins, amino acids, and minerals that play different biofunctional roles in the human metabolic system and as sources of nutrients for microbial growth (*Kantachote et al., 2016*). Probiotics are defined as "live microorganisms that, when administered in adequate amounts, confer a health benefit on the host" (*Hill et al., 2014*). Probiotic bacteria are used as nutritional supplements in functional foods,

which are easily digestible. Lactic acid bacteria are the most used for these purposes (*Plessas et al., 2017*). The health-improving properties of probiotics are associated with their positive influences on intestinal microbiota, intestinal functions, inhibition of the growth of pathogenic microorganisms, production of B-complex vitamins (especially folic acid), stimulation of the immune response, and reduction of symptoms of lactose intolerance (*Enujiugha & Badejo, 2017*). Despite coconut being associated with health benefits and used in culinary preparations, there are few studies investigating coconut extract in fermented beverages or desserts. Several authors have demonstrated that plant-based ingredients are suitable matrices for the growth of probiotics. These plant-based ingredients therefore enable the production of drinks and desserts with great market potential, due to innovation in functional products that use different processes and ingredients (*Lee, Boo & Liu, 2013*; *Prado et al., 2015*; *Kantachote et al., 2017*; *Zhang et al., 2018*). The main objective of this study was to develop a probiotic gelatin made from mature coconut ingredients (water and pulp). Thus, it is emphasized that dessert formulations associated with the nutritional value of coconut, using bacteria with probiotic potential, will lead to the development of a product that provides greater benefits to the consumer. This will result in differentiated products for the food sector, providing economic advantages, while also avoiding discards that contribute to environmental issues.

## MATERIALS AND METHODS

### Collection and characterization of samples

The present study used mature coconuts obtained from the rural area of the city of Lagoa Seca, Paraíba, Brazil. After harvesting, the coconuts were transported to the Food Research and Extension Center (NUPEA) at the State University of Paraíba, Campus I, located in the city of Campina Grande, Paraíba, Brazil. They were then washed for 10 min in a sodium hypochlorite solution, followed by a rinse with potable water to sanitize them. The washing was performed by placing the fruits in previously sanitized containers containing sodium hypochlorite diluted in water. For dilution, one tablespoon (10 ml) of sodium hypochlorite at a concentration of 2.5% was used for each liter of water. Subsequently, they were rinsed in potable water and stored. The coconut water was manually extracted by incising the fruits, making sure that there were no unusual appearances or odors. This odd appearance refers to dark spots or cracks, which may indicate that the inner pulp is spoiled. The obtained coconut water was passed through a 150 tyler/mesh sieve to remove any particles of coconut shell and fiber, then stored in containers and refrigerated at 10 ± 1 °C. Ripe fruits were selected based on their visual appearance, physical characteristics, and flowering time. At 7 months after flowering, the endosperm presents a firmer consistency, while at 12 months it is fully ripe, hardened, and with a darker color (*Rocha, Ferreira & Garcia, 2022*). The fruits were opened to obtain the pulp using a previously sanitized stainless steel knife. The pulp was manually removed and separated for extract preparation and characterization. After sanitization, both the coconut water and pulp were characterized for their mineral content (P, K, Ca, Mg, Na) using the *Tedesco et al. (1995)* methodology, pH using a digital pH meter (TECNAL, Piracicaba,

Brazil), and titratable acidity using volumetry with phenolphthalein indicator and 0.1 N sodium hydroxide solution (*Adolfo Lutz Institute, 2008*).

## Total phenolic compounds and antioxidant activity were also determined

Total phenolics were estimated according to the *Santos et al. (2017)* method using Folin-Ciocalteau reagent (Sigma-Aldrich Chamie GmbH, Steinheim, Germany). Antioxidant activity was determined through the DPPH assay using the *Rufino et al. (2007)* methodology. The coconut pulp was also characterized for its moisture, ash, protein, and lipid content according to the methodology of the *Adolfo Lutz Institute (2008)*, and its carbohydrate content was determined according to the *Food and Agriculture Organization of the United Nations: World Health Organization (2003)*.

## Preparation of the coconut vegetable extract

The coconut extract was prepared using mature coconut pulp and mature coconut water in a 1:1 ratio. The pulp, along with the water, was processed in a blender (Walita®, Philips, Brazil) at power level 2 for 3 min and strained to obtain the coconut extract.

## Inoculum preparation

The inoculum was prepared by reconstituting skim milk powder (Molico®, Nestlé Araçatuba, Brazil) at a concentration of 13 g $100 \text{ g}^{-1}$, which was then thermally treated by heating in a water bath at 85 °C for 15 min to eliminate vegetative forms of microorganisms and partially inactivate some enzymes. After cooling to 45 °C, a commercial probiotic culture of *Lacticaseibacillus rhamnosus* LR32 (LR32 200 B 100 GM, Floraft® Probiotics, Danisco USA Inc., Madison, WI, USA) was added at a ratio of $0.02 \text{ g mL}^{-1}$, followed by incubation at 37 °C for 2 h, following the methodology adapted from *Santos et al. (2019)*.

## Production of gelified product

The gelified product was produced using the extract, previously prepared, and mature coconut water in a ratio of 1:4. To this base, modified corn starch (Maizena®, Unilever, Garanhuns, Brazil), unsweetened gelatin powder (Dr. Oetker®, São Paulo, Brazil), and granulated sugar (Alegre®, Mamanguape, Brazil) at the following concentrations: maize starch 1.25%, gelatin powder 1.2%, sugar 6%. The mixture was processed in a blender (Walita®, Philips, Brazil) at power 2 for 1 min. The content was placed in an Erlenmeyer flask and heat treated at 85 °C for 15 min, cooled to 50–55 °C, inoculated with a previously prepared inoculum containing *L. rhamnosus* ($0.05 \text{ ml mL}^{-1}$), packaged in previously sterilized 130 g containers with lids, incubated in an oven at 37 °C for 2 h, and then refrigerated at 5 ± 2 °C for 21 days. The product was stored for evaluation of shelf life, texture profile analysis (firmness, adhesiveness, cohesiveness, elasticity, gumminess, and chewiness) was executed, acidity, and microbiological analysis (viability of probiotic culture) were performed on the day after production and on days 7, 14, and 21 from the production date.
### Analysis of texture profile

The texture profile of the product was evaluated in triplicate, following a methodology adapted from *Kusio et al. (2020)*, using a TA.XT Plus texture analyzer (Stable Micro Systems, Surrey, UK) with Exponent software version 4.0.6.0 (Stable Micro Systems, Surrey, UK).

### Determination of acidity

The titratable acidity was determined by volumetry with phenolphthalein indicator and 0.1 N sodium hydroxide solution. The values were expressed in terms of g of lactic acid per 100 g of sample (*Adolfo Lutz Institute, 2008*).

### Cell count determination

The enumeration of the probiotic microorganism population was carried out in triplicate, using the Pour Plate technique, with MRS agar (Man Rogosa & Sharpe) (Himedia, Maharashtra, India) acidified with acetic acid to pH 5.4, followed by incubation at 37 °C for 48 h. (*Silva et al., 2010*).

### Product characterization and centesimal composition

The product was also characterized for its moisture, ash, protein, and lipid content according to the methodology of the *Adolfo Lutz Institute (2008)* for coconut-based preparations. The analyses were performed in triplicate on the frozen product the day after preparation. The total carbohydrate content of the samples was obtained by difference (*Food and Agriculture Organization of the United Nations: World Health Organization, 2003*).

### Statistical analysis

The results were analyzed using the Statistica 7.0 software (Statsoft, Tulsa, OK, USA). Analysis of variance (ANOVA) was used to determine significant differences ($p < 0.05$) for each analyzed parameter between storage times. Differences were detected using the Tukey *post hoc* test.

## RESULTS

### Physicochemical characteristics of mature coconut water and mature coconut pulp

The fruit was characterized for pH and acidity measurement. The pH of the mature coconut water was 5.25 (Table 1). It varies according to the fruit's age; at 5 months, the pH is around 4.7 to 4.8, rising above five until the end of fruit growth (*Aragão, Isberner & Cruz, 2001*). *Prades et al. (2012)*, who studied the pH of different coconut waters, found values ranging from 5.1 to 6.1. The total acidity of the mature coconut water was 0.08 g of citric acid 100 g$^{-1}$ (Table 1) within the range determined by Brazilian legislation (*Brazil, 2009*), which establishes the minimum and maximum limits for acidity at 0.06% and 0.18%, respectively. Regarding the mineral content, the mature coconut water presented 19.57 mg L$^{-1}$ of sodium, 1,932.57 mg L$^{-1}$ of potassium, 279.93 mg L$^{-1}$ of calcium, 85.13 mg

**Table 1 The physicochemical parameters and mineral content obtained from the analysis of ripe coconut pulp and water.**

| Parameters | | Coconut water | Coconut pulp |
|---|---|---|---|
| pH | | 5.25 ± 0.02 | 7.35 ± 0.06 |
| Acidity (g of citric acid 100 g$^{-1}$) | | 0.08 ± 0.001 | 0.03 ± 0.02 |
| Mineral | Phosphorus | 11.17 | 3.32 |
| Coconut water (mg L$^{-1}$) | Magnesium | 85.13 | 2.18 |
| Coconut pulp (g kg$^{-1}$) | Sodium | 19.57 | 0.97 |
| | Potassium | 1,932.57 | 35.96 |
| | Calcium | 279.93 | 1.64 |

L$^{-1}$ of magnesium, and 11.17 mg L$^{-1}$ of phosphorus (Table 1). *Brito (2004)*, to produce a fruit nectar, characterized the mature coconut water of *C. nucifera* L. obtained from Itamaracá Island–PE, and obtained the following result in mineral analysis (mg%): 0.09 of phosphorus, 21.21 of magnesium, 23.60 of sodium, 125.50 of potassium, and 28.78 of calcium. In the study by *Kumar et al. (2021)*, which evaluated the changes in the chemical composition of coconut water during different stages of maturation of the Chowghat Orange Dwarf (COD) and Malayan Yellow Dwarf (MYD) coconut varieties, the following values were found for 10-month-old fruits (mg 100 ml-1): 8.2 of magnesium, 22.0 of sodium, 244.6 of potassium, and 14.2 of calcium. As for the mineral values of mature coconut pulp, phosphorus was 3.3 g kg$^{-1}$, magnesium was 22.18 g kg$^{-1}$, sodium was 0.97 g kg$^{-1}$, potassium was 35.96 g kg$^{-1}$, and calcium was 1.64 g kg$^{-1}$. According to the Brazilian Table of Food Composition (TACO), every 100 grams of mature coconut pulp has 117.5 mg of phosphorus, 51.5 mg of magnesium, 15.3 mg of sodium, 354.2 mg of potassium, and 6.5 mg of calcium. *Aroucha (2005)* verified this phenomenon when evaluating some chemical and physical characteristics of the fruit water from Anão, Verde, and Vermelho cultivars at different stages of maturation (mg 100 ml$^{-1}$). There was an increase in phosphorus up to the tenth month from 4.3 to 6.8 mg and a decrease to 3.7 mg in the twelfth month. On the other hand, magnesium and calcium decreased throughout ripening from 21.6 to 7.0 and 69.2 to 32.7 mg, respectively. *Medeiros et al. (2017)* analyzed the mineral content of *Cocos nucifera* L. pulp at different stages of maturation. The fruits were collected in the sixth, seventh, and eighth month after coconut inflorescence. The potassium content increased with ripening and showed significant differences among them using the Tukey test ($p < 0.05$). Potassium varied from 315.276 mg 100 g$^{-1}$ at 6 months, 506.263 mg 100 g$^{-1}$ at 7 months, and 618.874 mg 100 g$^{-1}$ at 8 months, being mineral with the highest quantity in the analysis. The main minerals in coconut sap present potassium levels (960.87 mg L$^{-1}$), sodium (183.21 mg L$^{-1}$), magnesium (22.91 mg L$^{-1}$), and calcium (0.42 mg L$^{-1}$) (*Asghar et al., 2020*). Total phenolic content and antioxidant capacity of water and pulp of mature coconut The water and pulp of mature coconut showed 5.72 and 9.77 GAE 100 mL$^{-1}$, respectively (Table 2). The total phenolic content reported by *Tan et al. (2014)* was 42.59 mg GAE L$^{-1}$ (4.259 GAE 100 mL$^{-1}$) in mature coconut water and ranged from 58.0 to 66.8 mg GAE L$^{-1}$ (5.80 to 6.68 mg GAE 100 mL$^{-1}$) also in mature

**Table 2 Total phenolics, EC$_{50}$ and antioxidant capacity of water and coconut pulp.**

| Parameters | Coconut water | Coconut pulp |
|---|---|---|
| Total phenolics (mg GAE 100 g$^{-1}$) | 5.72 ± 0.64 | 9.77 ± 1.63 |
| EC$_{50}$ (mg ml$^{-1}$) | 25.90 ± 1.78 | 15.28 ± 1.19 |
| Total antioxidant capacity (g product mg$^{-1}$ DPPH) | 1.67 ± 0.25 | 0.98 ± 0.16 |

coconut water in the study of *Mahayothee et al. (2016)*. The EC50 for water and pulp samples of mature coconut was 25.90 and 15.28 mg mL$^{-1}$, respectively (Table 2). The method and the form in which the samples are analyzed directly interfere with the results, as seen by *Rodsamran & Sothornvit (2018)* evaluating the antioxidant properties of mature coconut water with different solvents: Concentrated coconut water obtained by evaporation (14.03 mg mL$^{-1}$) extracted with acetone (14.34 mg mL$^{-1}$) and with diethyl ether (15.90 mg mL$^{-1}$). In the study by *Mohammed et al. (2021)*, the EC50 for coconut oil samples was in the range of 205.15–248.16 mg mL$^{-1}$, with lower radical activity due to the multiple stages applied for oil refining.

## Centesimal composition of mature coconut pulp

The centesimal composition of mature coconut pulp is shown in Table 3, with 53% moisture, 2.81% protein, 37.27% total lipids, 5.81% carbohydrates, and 1% ash. According to *TACO (2011)* data, the centesimal composition of mature coconut pulp, also known as copra, presents a moisture content of 43%, 3.7% protein, 42% total lipids, 10.4% carbohydrates, and 1% ash.

## Instrumental texture of the gelified product during shelf life

Table 4 presents the results of monitoring the texture profile parameters of the gelified product during a period of 21 days at 5 °C. There was a significant increase in firmness, adhesiveness, and gumminess until the 14th day ($p < 0.05$), followed by a significant reduction. The cohesiveness parameter did not present a significant difference between weeks, but there was a significant decrease from the 1st day to the 14th and 21st day ($p < 0.05$). Elasticity and chewiness on the 21st day were significantly reduced compared to the second week ($p < 0.05$), but not compared to the first week of storage. *Dokoohaki, Sekhavatizadeh & Hosseinzadeh (2019)* developed a dairy dessert containing microencapsulated *L. rhamnosus*. In this study, firmness, adhesiveness, gumminess, and chewiness increased during the storage period. Firmness increased by 54% for the dessert with free *L. rhamnosus*, 122% for the dessert with the microencapsulated form, and 31% for the control dessert. In another study, the firmness was 246.08 gf on the first day, increasing to 360.33 gf on the 21st day, and the gumminess was 127.61 and 457.70 gf (*Karimi, Sekhavatizadeh & Hosseinzadeh, 2021*), like those found in the present study (Table 4). The parameters were also like those of other studies with probiotic desserts. *Sousa et al. (2021)*, in their work on jabuticaba dairy dessert, found average cohesiveness values ranging from 0.49 to 0.59, values close to those obtained in this study (Table 4). Similar results were obtained in the banana gelified desserts of *Suebsaen et al. (2019)* with

**Table 3 Centesimal composition of ripe coconut pulp.**

| Parameters | Coconut pulp (%) |
|---|---|
| Moisture (g 100 g$^{-1}$) | 53 ± 0.04 |
| Ash (g 100 g$^{-1}$) | 1.11 ± 3.69 |
| Total lipids (g 100 g$^{-1}$) | 37.27 ± 1.71 |
| Protein (g 100 g$^{-1}$) | 2.81 ± 0.12 |
| Carbohydrates (g 100 g$^{-1}$) | 5.81 ± 0.00 |

**Table 4 Texture profile parameters of the probiotic gelatin created during its useful life.**

| Period (days) | Firmness (gf) | Adhesiveness (g.sec) | Cohesiveness | Elasticity (%) | Gumminess (gf) | Chewiness (gf) |
|---|---|---|---|---|---|---|
| 1 | 145.82 ± 2.75[a] | 692.85 ± 1.13[a] | 0.604 ± 0.02[b] | 90.1 ± 0.00[b] | 86.27 ± 0.08[a] | 77.72 ± 0.06[a] |
| 7 | 225.41 ± 2.36[c] | 1,185.81 ± 0.55[d] | 0.522 ± 0.02[b] | 91.4 ± 0.00[c] | 116.97 ± 2.47[c] | 106.76 ± 2.22[c] |
| 14 | 279.18 ± 1.04[d] | 1,042.96 ± 0.07[c] | 0.457 ± 0.01[ab] | 95.0 ± 0.00[d] | 122.45 ± 3.32[d] | 113.21 ± 1.52[d] |
| 21 | 206.81 ± 0.84[c] | 1,028.63 ± 0.70[b] | 0.473 ± 0.01[ab] | 88.1 ± 0.02[a] | 97.87 ± 2.51[b] | 91.981 ± 4.65[b] |

Notes:
Source: Prepared by the author, 2022.
Conventional signs used: a, b, c, d = Different superscript lowercase letters in the same column indicate significant differences between sampling periods ($p < 0.05$).

an elasticity of 72% in desserts containing gelatin in their formulation. A study conducted by *Morais (2014)*, evaluating the texture profile of a chocolate-flavored creamy dessert, found similar values to those obtained. It presented an average firmness ranging from 165.44 to 230.31 gf. It also obtained similar means, ranging from 105.32 to 176.25 gf, for the gumminess parameter.

## Microorganism viability and monitoring of acidity during shelf life

The average counts of *L. rhamnosus* in the desserts remained above 7 log CFU mL$^{-1}$ (Tables 5 and S1). throughout the studied storage period. *Almeida Neta et al. (2018)* observed that the viability of *L. rhamnosus* LR32 was also considerably stable in fermented milk dessert containing ingredients from *Plinia cauliflora* peel, remaining above 7 log CFU mL$^{-1}$ for 21 days. *Kantachote et al. (2017)* produced a potential probiotic beverage using coconut water supplemented with 0.5% monosodium glutamate and fermented with *Lactobacillus plantarum* DW12, presenting 8.4 log CFU mL$^{-1}$. Another study by *Lee, Boo & Liu (2013)* reported that after 2 days of fermentation of soft coconut water, probiotics (*Lactobacillus acidophilus* L10 and *Lactobacillus casei* L26) reached approximately 8 log CFU mL$^{-1}$. From the first day of storage, there was a significant increase in acidity levels ($p < 0.05$) and at the end of the 21-day shelf life, the dessert had an acidity of 0.64 g lactic acid 100 g$^{-1}$ (Tables 5 and S2).

## Compositional analysis of the gelified dessert

It is reported in the present study that the moisture content of the gelified dessert was 85.21% (Tables 6 and S3), which is similar to some authors who developed probiotic desserts. *Souza et al. (2021)* developed a creamy cupuaçu dessert and obtained a moisture value of 76.05%. *Corrêa, Castro & Saad (2008)* obtained values between 71.24% and

**Table 5 Viability of *L. rhamnosus* (log UFC g$^{-1}$) and acidity of the probiotic gelatin during the storage period of 21 days at 5 °C.**

| Period (days) | *L. rhamnosus* (log CFU g$^{-1}$) | Acidity (g lactic acid 100 g$^{-1}$) |
|---|---|---|
| 0 | 7.49 ± 0.09[a] | 0.15 ± 0.03[a] |
| 1 | 7.73 ± 0.03[b] | 0.15 ± 0.02[b] |
| 7 | 8.67 ± 0.09[b] | 0.43 ± 0.02[c] |
| 14 | 8.55 ± 0.01[c] | 0.48 ± 0.01[d] |
| 21 | 8.51 ± 0.02[c] | 0.64 ± 0.01[d] |

Notes:
Source: Prepared by the author, 2022.
Conventional signs used: a, b, c, d = Different superscript lowercase letters in the same column indicate significant differences between sampling periods ($p < 0.05$).

**Table 6 Centesimal composition of the probiotic gelatina.**

| Parameters | Values (%) |
|---|---|
| Moisture (g 100 g$^{-1}$) | 85.21 ± 0.05 |
| Ash (g 100 g$^{-1}$) | 0.54 ± 0.018 |
| Total lipids (g 100 g$^{-1}$) | 4.88 ± 0.33 |
| Protein (g 100 g$^{-1}$) | 4 ± 0.00 |
| Carbohydrates (g 100 g$^{-1}$) | 5.37 ± 0.00 |

71.67% in their probiotic coconut flan. Regarding the protein content, the value of 4.88% (Tables 6 and S4) was close to the vegetable coconut mousse dessert (4.4%) developed by *Kobus-Cisowska et al. (2021)* with the addition of almond protein and fruit puree. As for the lipid content, high values are found in other coconut-based products, such as in the study by *Lima et al. (2018)* who produced lactose-free yogurt using coconut milk as a base and found 6.83g 100 g$^{-1}$ of lipids. The values found are within the expected range for fruit-based products, mainly composed of carbohydrates. The carbohydrate content of 5.37% (Table 6) was higher than that found by *Lima et al. (2018)* in coconut milk-based yogurt (3.23%) and lower than that found in coconut mousse (12.1%) (*Kobus-Cisowska et al., 2021*).

## DISCUSSION

### Physicochemical characterization of the pulp and water of mature coconut

The increase in pH with maturity correlates well with the concomitant decrease in titratable acidity. The decrease in acidity may be due to the reduction in the number of organic acids and ascorbic acid present in the water with maturity (*Kumar et al., 2021*). The study by *Costa et al. (2021)*, aiming to apply food byproducts in the development of efficient new functional foods and beverages, used mature coconut water which presented a titratable acidity of 0.14 ± 0.01 g of citric acid 100 mL$^{-1}$, a value close to those found in the present study (Table 1). *Imaizumi et al. (2016)* found highly varied values of titratable acidity (0.23% to 0.96%) for in natura coconut water from different cities in Brazil.

Coconut water has a natural balance of sodium, potassium, calcium, and magnesium, which makes it a very healthy electrolytic drink. The results of the minerals differed from other studies due to the variation in planting locations and, consequently, the soil composition that directly affects the mineral content of the fruit, both pulp and water. The present study was carried out with coconuts from an inland city, a non-coastal region. Most studies that analyze the composition of coconut water do not inform the planting location of the fruit, and when they do, it is a coastal region (*Vigliar, Sdepanian & Fagundes-Neto, 2006*). In addition, it is seen that there is a variation in the content of minerals during plant development; studies show a peak at 10 months of age and a decline at 12 months for some minerals, while others remain and some even increase at the end of maturation (*Aroucha, 2005*; *Brito, 2004*).

## Total phenolics and antioxidant capacity of mature coconut pulp and water

The antioxidant capacity varies according to the method employed, the type of solvent, and the form in which the raw material is available (*Rodsamran & Sothornvit, 2018*). *Mahayothee et al. (2016)* investigated the antioxidant activities and identified the existing phenolic compounds in the water and solid endosperm of coconut at three different stages of maturity. The total phenolic content and antioxidant activity indices increased with coconut maturity, and the amounts of phenolic compounds found in the water were lower than those in the solid endosperm, as in the present study (Table 2). The DPPH technique with stable organic radical 1.1-diphenyl-2 picrylhydrazyl was used to determine free radical scavenging activity. It was described as the number of antioxidants needed to decrease the initial concentration of DPPH by 50%. Thus, the lower the EC50 value, the higher the antioxidant activity. This trial is based on the colorimetric change that DPPH undergoes in the presence of an antioxidant. It can be quantified by measuring its absorbance, where a decrease correlates with the DPPH elimination activity of the antioxidant (*Mohammed et al., 2021*).

## The centesimal composition of mature coconut pulp

The centesimal composition of mature coconut pulp was similar to that described in the literature, with slightly higher moisture content and lower carbohydrate content (Table 3). The variation in macronutrient content of the fruits can be explained by the influence of environmental factors, justified by the geographical characteristics of each place, such as rainfall, temperature, humidity, and soil fertility, by the time and region of harvest, or by the cultivar analyzed (*Soares et al., 2015*).

## The texture of dessert during shelf life

The formation and texture of gels are influenced by several factors such as acidity, temperature, the presence of metal ions, and solutes such as sugars in the mixture, highlighting the importance of monitoring the textural parameters of desserts (*Banerjee & Bhattacharya, 2012*). The perception of food is not only characterized by factors such as smell, taste, and texture sensations but rather the interaction of all these

factors, combined with nutritional value, these contribute to the image and quality conveyed to the consumer. Texture definitions are important in understanding the effects that each ingredient has on a particular food. Texture analysis in food allows decision-making regarding the limit quantity of inputs to be used, to obtain the same response with maximum economy, optimizing the entire process and/or product (*Dokoohaki, Sekhavatizadeh & Hosseinzadeh, 2019*). Regarding the texture profile, adhesiveness, gumminess, firmness, and, directly, chewiness increased during the storage period (Table 4). This increase is a process that occurs after the storage and refrigeration of gelatinized starch in which the interactions of the polysaccharide molecules with the gel water lose energy, strengthening the hydrogen bonds between the hydroxyls of the glycosidic units. Starch chains re-associate to form crystalline areas (*Paramasivam et al., 2021*). The results of firmness and gumminess exhibit a correlation, meaning that with a decrease in firmness, there will consequently be a decrease in gumminess. Regarding the change in firmness and adhesiveness after 14 days, this alteration in molecular structure may stem from the proteolytic action of the microorganism (*Salles et al., 2022*) or from the increase in acidity throughout storage due to the metabolism of *L. rhamnosus*. In the development of umbu-cajá jelly, *Oliveira et al. (2014)* observed that the titratable acidity of the formulated product was around 0.8%, close to the value found in this study (Table 5), and did not exhibit syneresis.

## Probiotic microorganism viability

The average counts of *L. rhamnosus* in the desserts remained above 7 log CFU ml$^{-1}$ throughout the storage period studied (Table 5). It is widely accepted that at least 106 to 107 viable probiotic cells must be present in the final product at the time of consumption. Regarding the matrix, a study by *Mauro & Garcia (2019)* investigated the survival of two strains of *Lactobacillus reuteri*, LR 92 and DSM 17938, in coconut beverages. The un-supplemented coconut extract provided sufficient substrate for *L. reuteri* growth, due to the natural sucrose present in the matrix used as a source for bacterial multiplication. In this research, no reduction in viability was observed during storage, despite the increase in acidity, corroborating with several studies that reported the maintenance of the minimum viable amount of probiotic culture during storage (*Aragon-Alegro et al., 2007*; *Corrêa, Castro & Saad, 2008*; *Silva et al., 2012*). The concentration of titratable acidity in the products is dependent on the raw material acid and the production of organic acids by the probiotic microorganism (*Lobato-Calleros et al., 2014*). In addition, the behavior of probiotic bacteria in a food matrix will depend on the characteristics of each species (*Tripathi & Giri, 2014*). The species *L. rhamnosus* is represented by heterofermentative, facultative strains that ferment hexoses such as lactose and fructose into lactic acid, and pentoses into a mixture of lactic and acetic acid (*Silva et al., 2013*; *Hammes & Vogel, 1995*). The variation from the first to the twenty-first day (post-acidification) for both formulations indicates that fermentation occurred (Table 5). The initially low values correspond to the culture adaptation period to the matrix. Throughout the storage period, lactic acid bacteria use the substrates present in food for metabolic activities resulting in the production of organic acids such as lactic acid (*Nakkarach & Withayagiat, 2018*).

### Centesimal composition of the product

The coconut-based formulation presented centesimal composition values within the expected range for fruit-based products (Table 6). There is currently no legislation regarding vegetable desserts or those using fruits as a base for production. The coconut base becomes an interesting option to improve the benefits of dairy desserts. It has a white color as does milk and a characteristic aroma that pleases consumers, as sensory quality is a critical determinant in the choice of a product (*Patil & Benjakul, 2018*). While cow's milk has a lipid content close to 3%, coconut milk is around 24%, which contributes to the increase in the product's lipid content but also to the enrichment of its nutritional value, as it is mostly composed of medium-chain fatty acids. These fatty acids have beneficial effects on the body, such as increasing basal metabolism and aiding in the absorption of vitamins and minerals (*Patil & Benjakul, 2018*).

## CONCLUSIONS

The physical and chemical characteristics of coconut water and pulp, as well as the mineral content determination, showed values compatible with those of ripe fruits according to literature data. With ripe coconut water and pulp, it was possible to obtain a differentiated product that contributes to diversification in the food sector. The addition of probiotic microorganisms to the dessert with vegetable ingredient produced in this study added greater functionality to the product, because of the improvement in nutritional, sensory, and food safety characteristics. The post-acidification phenomenon observed was compatible with the type of microorganism used, and it remained viable throughout the storage period. The texture profile during the 21 days varied significantly. There was a decrease in cohesiveness, but without the presence of syneresis, and an increase in the other parameters (firmness, adhesiveness, elasticity, gumminess, and chewiness) due to the components of the formulation, a result also observed in other studies of desserts with starch.

The developed product has a centesimal composition like that of other probiotic dessert studies, with a predominance of carbohydrates due to starch and sugar. It represents a viable alternative for maintaining consumer health as it is a source of nutrients and bioactive compounds, as well as providing cost reduction in inputs compared to desserts already available on the market. To cater to a larger consumer audience and address some limitations of the current study, a version of the gelatin substitute replacing sugar with sweetener would be an option for diabetics and individuals on calorie-deficit diets. Additionally, in order to avoid post-acidification, the addition of prebiotics or even microencapsulation of the probiotic culture in this matrix would be an alternative for maintaining the number of viable cells in quantities recommended by Brazilian legislation with moderate acidity.

## ACKNOWLEDGEMENTS

To the Graduate Program in Pharmaceutical Sciences and to the Coordination for the Improvement of Higher Education Personnel (CAPES). To the Food Research and Extension Center (NUPEA). To the State University of Paraíba (UEPB) and to the staff,

technicians, and other collaborators of UEPB. The authors thank the Danisco, which provided probiotic strains for this study.

### Funding

This work was supported by the Coordenação de Aperfeiçoamento de Pessoal de Nível Superior: CAPES/PROAP Fundação Parque Tecnológico da Paraíba: PaqTcPB, Project AC-NUPEA-UEPB; Fundação de Apoio à Pesquisa do Estado da Paraíba: FAPESQ, Project 028/2018; Conselho Nacional de Desenvolvimento Científico e Tecnológico: CNPq, Project 307075/2020-6; Universidade Estadual da Paraíba: Grant #01/2024 PRPGP/UEPB. The funders had no role in study design, data collection and analysis, decision to publish, or preparation of the manuscript.

### Grant Disclosures

The following grant information was disclosed by the authors:
Coordenação de Aperfeiçoamento de Pessoal de Nível Superior: CAPES/PROAP
Fundação Parque Tecnológico da Paraíba: PaqTcPB, Project AC-NUPEA-UEPB.
Fundação de Apoio à Pesquisa do Estado da Paraíba: FAPESQ, Project 028/2018.
Conselho Nacional de Desenvolvimento Científico e Tecnológico: CNPq, Project 307075/2020-6.
Universidade Estadual da Paraíba: #01/2024 PRPGP/UEPB.

### Competing Interests

The authors declare that they have no competing interests.

### Author Contributions

- Beatriz Patricio Rocha conceived and designed the experiments, performed the experiments, analyzed the data, prepared figures and/or tables, authored or reviewed drafts of the article, and approved the final draft.
- Pedro Luan de Brito Lopes conceived and designed the experiments, performed the experiments, authored or reviewed drafts of the article, and approved the final draft.
- Miqueas Oliveira Morais da Silva analyzed the data, prepared figures and/or tables, authored or reviewed drafts of the article, and approved the final draft.
- Ana Catarina Guimarães Gomes analyzed the data, prepared figures and/or tables, authored or reviewed drafts of the article, and approved the final draft.
- Flá*via* Carolina Alonso Buriti analyzed the data, authored or reviewed drafts of the article, and approved the final draft.
- Isanna Menezes Florêncio performed the experiments, authored or reviewed drafts of the article, and approved the final draft.
- Eliane Rolim Florentino conceived and designed the experiments, performed the experiments, analyzed the data, authored or reviewed drafts of the article, and approved the final draft.

## Data Availability

Raw measurements are available in the Supplemental Files.

## Supplemental Information

Supplemental information for this article can be found online at http://dx.doi.org/10.7717/peerj.17502#supplemental-information.

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
