# Peer review of "Utilization of ripe coconut water in the development of probiotic gelatin"

_PeerJ, doi:10.7717/peerj.17502_

## Round 0.1 · original submission · Major Revisions

Please, authors, kindly address the comments raised by the reviewers. Please make an effort to provide thorough detail in your replies. Look forward to your revised manuscript.

**Language Note:** The review process has identified that the English language must be improved. PeerJ can provide language editing services - please contact us at [email protected] for pricing (be sure to provide your manuscript number and title). Alternatively, you should make your own arrangements to improve the language quality and provide details in your response letter. – PeerJ Staff

Reviewer 1 ·

Basic reporting

In general, the work is innovative and fairly carried out.
However, some of the units need to be corrected and Tables require further editing to improve the quality of prrsentation.
Physical-chemical should be physicochemical
Use . (full stop/point) to indicate decimal place not , (comma). For eg. 5,81±0,00 should be 5.81±0.00. Use consistent number of decimal place for alll values in a given Table.
For example, Line 189-190: mgL-1 should be mg/L or mgLsuperscript-1.

Experimental design

The methods are acceptable and were fairly done.

Validity of the findings

Table 1 is scanty and unneccessary. This could just be stated in text.
Combine Tables 2 and 3 into one.
Table 6: Texture profile of what?
Table 7: Specify the product
Table 8: what product?

Additional comments

It will be helful to provide cross-reference to the specific Tables/results in the Discussion section.

Reviewer 2 ·

Basic reporting

In the manuscript, under the section "Physical-chemical characteristics of water and mature coconut pulp" the notations used for the concentration, specifically "L-1" and "g-1," might benefit from a slight clarification to enhance readability and precision. It would be advisable to denote these units with the "-1" as an exponent, signifying that it represents an inverse or reciprocal measure. Adopting this convention ensures that the readership immediately understands the units without necessitating further interpretation.

Experimental design

In the manuscript, under the section titled "Production of Gelified Product," the choice of a 1:4 ratio mentioned on line 140 could be justified. It would benefit the readers if the authors could provide a rationale for selecting this specific ratio for preparing the gelified product. Additionally, it would be informative to know whether other ratios were considered or experimented with during the study. If so, the outcomes or reasons for not choosing them could be elaborated upon. This information could significantly enhance the understanding of the methodological choices made in the study.

Validity of the findings

The manuscript comprehensively provides all necessary underlying data, which are demonstrated to be robust, statistically sound, and well-controlled. The conclusions drawn are articulated, directly tied to the original research question, and appropriately confined to the results that support them. This thorough approach ensures the reliability and validity of the findings presented.

Additional comments

The authors have developed a probiotic gelatin utilizing pulp and water derived from mature coconuts. This innovative approach has been thoroughly evaluated for its physicochemical properties alongside the viability of the Lacticaseibacillus rhamnosus LR32 strain within the medium. Additionally, the texture properties of the resultant product have been rigorously addressed. The comprehensive examination of these aspects contributes significantly to understanding this novel probiotic gelatin's potential applications and benefits. The approach is novel.

Reviewer 3 ·

Basic reporting

The English proficiency is good and meets the standard for publication. Adequate literature has been cited showing that the author understands the coconut industry. The structure is appropriate based on the format of the Journal.

The Manuscript meet the basic reporting format

Experimental design

Generally good, however, note take of the observations below and respond appropriately;

1. What was the state of the maturity of the coconut? Maturity influences physicochemical attributes, functional and nutritional characteristics
2. Describe how and conditions under which the coconut fruits were transported to the research centre. The condition is key
3. State the concentration of Sodium hypochlorite used
4. How was the washing done for the 10 minutes?
5. Describe the manual extraction process of the coconut water. It is important in the microbial and safety of the final products
6. What is unusual appearance (line 107)? not clear. Describe it well
7. Can you propose how the by-product, following the preparation of the coconut vegetable extract, should be utilised to prevent waste in the coconut processing industry?
8. Line 145 rather should be constructed as : .... inoculated with a previously prepared inoculum containing L. rhamnosus
9. 172: Why is it Capitalised
10. 171: should be frozen product

Validity of the findings

The data presented is good enough to warrant publication and also buttresses the gap and objective projected.

Observations for attention;
1. 192: The Scientific name should be italicised or underlined
2. Ensure consistency in the use of symbols to represent a decimal. In text, it was dot (.) but in tables it was common (,).
3. line 293: The increase in pH with maturity correlates…. However, your work did not determine the effect of maturity on pH.
4. Table 7. The presentation of significant differences in the table is incorrect. rectify
5. The conclusion was more of a summary of the findings, however, it should throw light on the objective/research questions rather. Reconsider

Additional comments

General comments
1. Title: inclusion of Industry is not appropriate, hence may delete or restructure the title
2. Introduction:
1. Coconut serving as a source of nutrients to support the growth of microorganisms brings the concept of a prebiotic. Hence the project may focus on the prebiotic potential of coconut
2. The last sentence is too long and not clear. Restructure to make it meaningful

Reviewer 4 ·

Basic reporting

Considered by the topic, the manuscript is quite suitable for this journal. It already stated the novelty of the research the fill the research gap in this kind of study.

To improve this manuscript, there are several points that shall be revised and improved. We state all the points below.

Experimental design

The topic of utilizing by-product to develop sustainable functional foods is very interesting and relatable to the environment and industrial problem nowadays. By far, this research was done quite comprehensively with many basic test methodology involved.

Validity of the findings

1. The usage of ‘byproduct’ word either on the title or in the overall manuscript should be consistent. ‘byproduct’ or ‘by-product’

2. The usage of ‘physical-chemical’ word in the overall manuscript should be consistent. ‘physical-chemical’ or ‘physicochemical’

3. In Abstract section (background, line 25) please add ‘coconut pulp’ after ‘coconut water’ to become more consistent throughout the whole manuscript.

4. In the introduction, please add some short explanation about coconut pulp, for example: the explanation of coconut pulp definition and why the coconut pulp could become a byproduct? – to complement the explanation of coconut water which has already good and on point.

5. In the Materials & Methods section (line 110), please add some explanation of how to determine mature visual appearance, for example by using color index, or maturity time after flowering, etc.

6. In the Materials & Methods section (line 111), please add some explanation about the method of how to obtain or make the coconut pulp.

7. In the Discussion section, please add some explanation about why the authors used 1:1 ratio of coconut water and coconut pulp (line 124), instead of other ratio such as 1:2 or 2:1.

8. In the Discussion section of phenolic and antioxidant potency (line 311 and below), please give some explanation of why phenolic content of pulp is greater than coconut water; vice versa why antioxidant activity of coconut water is greater than coconut pulp.

9. in the Discussion section of texture (line 333 and below), please give explanation about the cause of the decrease of firmness, adhesiveness, and gumminess above 14 days storage time. Maybe the cause is water syneresis of the reduce of water holding capacity, etc.

10. In the Discussion section (and subsection), please add or cite the previous authors the have been used as a reference. For example in line 291: ‘The pH of coconut water was compatible with results from other authors.’; add the name of the authors (Aaaaa et al., 2xxx). This statement shall be applied thoroughly through out the overall manuscript; for example: line 312, 327, etc.

Additional comments

1. The consitency of writing measurement units shall be given more attention. Either (mg mL-1) or (mg/mL), it shall be very consistent throughout the whole manuscript to avoid confusion.

2. The writing of Latin names shall be given more attention throughout the whole manuscript.

3. The consistency of writing reference citation in the manuscript. Either (Aaaaa et al., 2xxx) or (AAAAAA et al., 2xxxx). Please revise it according to the journal format.

4. Please give more recent publication as a reference for this manuscript.

---

## Round 0.2 · Minor Revisions

Thank you authors for your kind patience as reviewers considered your work.
Please, you can find that the reviewers considered your work very favorably, some have raised some comments, which deserve your careful attention.
Kindly attend to them diligently...remember, clarity is very important. Also, kindly look at the annotated manuscript attached, and carefully address the comments.
Look forward to your revised manuscript.

Reviewer 1 ·

Basic reporting

No comment

Experimental design

No comment

Validity of the findings

No comment

Additional comments

The revised paper is acceptable. However, the revised file reviewed appears distorted and should be formatted properly.

Reviewer 2 ·

Basic reporting

The latest draft of the manuscript exhibits a commendable use of clear and professional language, ensuring an articulate presentation of its contents. It thoughtfully introduces relevant hypotheses, which are effectively supported by well-substantiated results. This approach enhances the document's credibility and facilitates a deeper understanding of its thematic undertakings.

Experimental design

The experimental design is relevant, well-detailed, and meaningful. The design holds significant meaning, as it is tailored to address the core questions and hypotheses of the study.

Validity of the findings

The conclusions drawn from the study align closely with the outcomes observed indicating that the analysis conducted was accurate in capturing the essence of the data.

Reviewer 3 ·

Basic reporting

The English proficiency is fairly OK. Pertinent literature has been cited showing that the author understands the coconut industry.

Experimental design

Generally good

Validity of the findings

Data as referenced in the main text have been provided and well presented

Additional comments

1. 157. The word “bases” may not be appropriate
2. Restructure the sentence: 157-163. It is too long and not clear
3. May delete …… “using this byproduct” from the last sentence under the introduction
4. Line 267: Rephrase the Subheading: Physicochemical characteristics of water and mature coconut pulp. The question is which water ??

Reviewer 4 ·

Basic reporting

The manuscript has shown a great improvement on its typology and format that shall meet the journal needs. After thoroughly read the manuscript, many substantial reasons have already been explained well enough to meet the publication standard. All definition of the materials and methods have also been explained thoroughly. Therefore, I highly recommend this manuscript to be accepted in this publication.

Thank you.

Experimental design

No comment

(The experimental design has already met the journal requirement, and all the methods used in the research were quite comprehensive)

Validity of the findings

No comment

(The research results have already showed good novelty in this subject area)

Additional comments

Basically, after a thorough revision, the research is already proper to be published.

Annotated reviews are not available for download in order to protect the identity of reviewers who chose to remain anonymous.

---

## Round 0.3 · accepted · Accept

I have checked this revised version. I am very convinced it is acceptable for publication. Thank you authors for finding PeerJ as your choice journal, and look forward to your future scholarly contributions. Congratulations